# Antibacterial Activity of Amidodithiophosphonato Nickel(II) Complexes: An Experimental and Theoretical Approach

**DOI:** 10.3390/molecules25092052

**Published:** 2020-04-28

**Authors:** Enrico Podda, Massimiliano Arca, Giulia Atzeni, Simon J. Coles, Antonella Ibba, Francesco Isaia, Vito Lippolis, Germano Orrù, James B. Orton, Anna Pintus, Enrica Tuveri, M. Carla Aragoni

**Affiliations:** 1Dipartimento di Scienze Chimiche e Geologiche, Università degli Studi di Cagliari, Cittadella Universitaria, SS. 554 bivio Sestu, 09042 Monserrato–Cagliari, Italy; 2UK National Crystallography Service, School of Chemistry, Faculty of Engineering and Physical Sciences, University of Southampton, Southampton SO17 1BJ, UK; 3Department of Surgical Sciences, University of Cagliari, 09042 Cagliari, Italy; 4National Research Council of Italy, ISPA-CNR, 07100 Sassari, Italy

**Keywords:** amidodithiophosphonate, nickel complexes, antibacterial activity, density functional theory (DFT), X-ray diffraction, P–N cleavage

## Abstract

The reactions of 2,4-bis(4-methoxyphenyl)-1,3-dithio-2,4-diphosphetane-2,4-disulfide (Lawesson’s Reagent, LR) with benzylamine (BzNH_2_) and 4-phenylbutylamine (PhBuNH_2_) yield benzylammonium *P*-(4-methoxyphenyl)-*N*-benzyl-amidodithiophosphonate (BzNH_3_)(BzNH-adtp) and 4-phenylbutylammonium *P*-(4-methoxyphenyl)-*N*-(4-phenylbutyl)-amidodithiophosphonate (PhBuNH_3_)(PhBuNH-adtp). The relevant nickel complexes [Ni(BzNH-adtp)_2_] and [Ni(PhBuNH-adtp)_2_] and the corresponding hydrolysed derivatives (BzNH_3_)_2_[Ni(dtp)_2_] and (PhBuNH_3_)_2_[Ni(dtp)_2_] were prepared and fully characterized. The antimicrobial activity of the aforementioned amidodithiophosphonates against a set of Gram-positive and Gram-negative pathogen bacteria was evaluated, and [Ni(BzNH-adtp)_2_] and [Ni(PhBuNH-adtp)_2_] showed antiproliferative activity towards *Staphylococcus aureus* and *Staphylococcus haemolyticus* strains. density functional theory (DFT) calculations were performed to shed some light on the activity of reported compounds related to their tendency towards P–N bond cleavage.

## 1. Introduction

Phosphorus-1,1-dithiolates such as dithiophosphates, dithiophosphinates, dithiophosphonates, and amidodithiophosphonates (**I**, **II**, **III** and **IV**, respectively, see Scheme 1), are important classes of sulfur-donor anionic ligands that display a multiplicity of coordination patterns with transition metal ions and main group elements [1,2,3]. The PS_2_^−^ moiety can coordinate in monodentate, bidentate (with either symmetric or asymmetric bonding), and polydentate modes. A huge variety of both discrete and polymeric structures are prevalent in the literature [4]. Since the early 1960s, dithiophosphates and dithiophosphinates gained increasing importance due to their applications as pesticides and extracting agents in mineral ores, and a large amount of information describing the reactivity of compounds **I** and **II** was reported [5]. On the contrary, due to synthetic difficulties, the chemistry of dithiophosphonate complexes (**III**) became relevant only after the turn of the new century, with the development of a novel synthetic route starting from 1,3-dithiadiphosphetane-2,4-disulfides (such as Lawesson’s Reagent) [3,6]. A similar synthetic route was used to prepare amidodithiophosphonates (**IV** in Scheme 1), a class of phosphorus-1,1-dithiolates, featuring a P–N bond, that still remains largely unexplored. 

The tendency to undergo cleavage of the P–N bond to give the corresponding dithiophosphonic acid may explain the low occurrence of structurally characterized amidodithiophosphonates (Figure 1). The majority of known structures are in an anionic form, with the released protonated amine acting as a counterion [7,8,9,10].

Some examples are also reported where the complete hydrolysis of P–N and P–S bonds in amidodithiophosphonates yields phosphonates, with a concurrent loss of the amine and hydrogen sulphide [11,12]. 

Recently, the antiproliferative and antibacterial activity of this class of compounds was reported to be related to the slow release of H_2_S [11,12]. This aspect is of particular interest as antimicrobial resistance is becoming one of the principal public health problems of the 21st century [13,14,15]. In the search for novel antimicrobial agents, coordination compounds containing transition metal ions represent a promising avenue for drug development [16,17]. Complexes of metals such as Au, Ir, Co, and Cu have demonstrated an excellent activity against aerobic Gram-positive pathogenic bacteria, such as *Staphylococcus* spp. [17,18,19,20]. This bacterial group, in particular *S. aureus*, can cause many forms of infections in different organs and is one of the major causes of nosocomial infections of surgical wounds and in indwelling medical devices [21,22]. Methicillin-resistant *S. aureus* (MRSA) is solely responsible for many life-threatening nosocomial infections in humans, causing an increase both in the treatment duration and medical costs [23]. The problem of resistance is amplified by the ability of *S. aureus* to form biofilms on biotic and abiotic surfaces and is of particular concern with several implanted medical devices [24,25,26]. Bacteria in these biofilms are stubbornly difficult to treat because such microbial aggregates are traditionally considered impervious to drug diffusion [27,28]. 

Given the scarcity of data reported on square-planar complexes of d^8^ metal ions with potential antimycotic and antimicrobial activity [3,11,12,18,29,30] and the different hydrolytic products described [7,8,9,10,11,12], we report here the synthesis, characterization and activity (against a set of Gram-positive and Gram-negative pathogenic bacteria), of the novel benzylammonium *P*-(4-methoxyphenyl)-*N*-benzyl-amidodithiophosphonate (BzNH_3_)(BzNH-adtp), 4-phenylbutylammonium *P*-(4-methoxyphenyl)-*N*-(4-phenylbutyl)-amidodithiophosphonate (PhBuNH_3_)(PhBuNH-adtp), their nickel complexes [Ni(BzNH-adtp)_2_] and [Ni(PhBuNH-adtp)_2_] and the corresponding hydrolytic products (BzNH_3_)_2_[Ni(dtp)_2_] and (PhBuNH_3_)_2_[Ni(dtp)_2_] (Scheme 2).

## 2. Results

The reaction between 2,4-bis(4-methoxyphenyl)-1,3,2,4-dithiadiphosphetane-2,4-disulfide (Lawesson’s Reagent, LR) and primary amines ^i^PrNH_2_, BzNH_2,_ and PhBuNH_2_ (isopropylamine, benzylamine, and 4-phenylbutylamine, respectively) in toluene gave rise to the corresponding amidodithiophosphonate ammonium salts (^i^PrNH_3_)(^i^PrNH-adtp), (BzNH_3_)(BzNH-adtp), and (PhBuNH_3_)(PhBuNH-adtp) (Scheme 2). The compounds were characterized by m.p. determination, FT-IR, ^1^H-NMR, and ^31^P{^1^H} NMR spectroscopies (see experimental for details). The FT-IR spectra of (BzNH_3_)(BzNH-adtp) and (PhBuNH_3_)(PhBuNH-adtp) (Appendix A) show the N–H stretching frequency as a strong single peak at around 3300 cm^−1^. The bands peculiar to the asymmetric and symmetric P–S stretching vibrations can be envisaged at about 624 and 550 cm^−1^, respectively [30]. The ^1^H-NMR spectra of (BzNH_3_)(BzNH-adtp) and (PhBuNH_3_)(PhBuNH-adtp) in DMSO-d_6_ solutions are shown in Appendix A and are consistent with those reported for the amidodithiophosphonate salt prepared from benzylamine and phenylethylamine and LR, corroborating the formation of the P–N bond and the proposed structure [31,32]. The ^31^P{^1^H} NMR spectra of (BzNH_3_)(BzNH-adtp) and (PhBuNH_3_)(PhBuNH-adtp) show a singlet at 79.7 and 93.1 ppm, respectively, in agreement with what it was previously reported for (^i^PrNH_3_)(^i^PrNH-adtp) and similar known compounds [7,8,9,10,30,31,32]. 

The reaction of the amidodithiophosphonate salts with NiCl_2_·6H_2_O afforded the corresponding nickel(II) complexes [Ni(^i^PrNH-adtp)_2_], [Ni(BzNH-adtp)_2_] and [Ni(PhBuNH-adtp)_2_] (Scheme 2), as purple solids. The obtained compounds were fully characterized by elemental analysis, m.p. determination, FT-IR and ^1^H and ^31^P{^1^H} NMR spectroscopies, confirming their nature of amidodithiophosphonato nickel complexes. The FT-IR spectra of the compounds (Appendix A) show the N–H stretching vibration as a strong single peak falling at around 3250 cm^−1^, and the bands for asymmetric and symmetric P–S stretching modes are found, as expected, around 660 and 560 cm^−1^, respectively [29]. Due to the peak broadening encountered in DMSO-*d*_6_ solutions, ^1^H-NMR spectra of [Ni(BzNH-adtp)_2_] and [Ni(PhBuNH-adtp)_2_] were recorded in CDCl_3_ solution. ^1^H-NMR data are consistent with the proposed structure (Scheme 2). In particular, the ^1^H-NMR spectra of [Ni(BzNH-adtp)_2_] (Appendix A) show three signals in the aliphatic region at 3.34–3.20, 3.86, and 4.35–4.21 ppm, assigned to the protons of NH, OCH_3_ and CH_2_ moieties, respectively. The aromatic portion of the spectrum displays two signals at 8.04–7.87 and 7.05–6.93 ppm, attributed to the protons of the aryl ring directly bonded to the phosphorus atom. The amine aromatic protons are observed at 7.40–7.16 ppm. The ^1^H-NMR spectrum of [Ni(PhBuNH-adtp)_2_] (Appendix A) shows the signal that can be assigned to CH_2_ protons overlapped with the water residue; the signals at 2.62, and 3.17–2.99 ppm can be attributed to the protons of the aliphatic chain of the amine, and the broad signal at around 2.91 ppm can be assigned to the NH proton, similar to that found for (BzNH_3_)(BzNH-adtp) and [Ni(BzNH-adtp)_2_]; the singlet at 3.86 ppm is attributed to the protons of the OCH_3_ group. The protons of the aromatic portion display signals at 7.90 and 6.97 ppm, assigned to the methoxyphenyl P-substituent, and signals between 7.3–7.1 ppm can be assigned to the aromatic protons of the amine. The ^31^P{^1^H} NMR spectra of [Ni(BzNH-adtp)_2_] and [Ni(PhBuNH-adtp)_2_] recorded in DMSO-d_6_ show only a singlet at 75.8 and 76.2 ppm, respectively, and are comparable with the CP-MAS ^31^P value of 71.7 ppm measured for of [Ni(^i^PrNH-adtp)_2_] and consistent with the presence of the P–N bond.

The attempts of growing single crystals suitable for X-ray analysis for [Ni(BzNH-adtp)_2_] and [Ni(PhBuNH-adtp)_2_] from either acetonitrile or chloroform solutions, revealed that hydrolysis occurred, resulting in the formation of compounds (BzNH_3_)_2_[Ni(dtp)_2_]·2H_2_O and (PhBuNH_3_)_2_[Ni(dtp)_2_], featuring the anionic dithiophosphonato Ni complex [Ni(dtp)_2_]^2–^ counterbalanced by two BzNH_3_^+^ and PhBuNH_3_^+^ cations, respectively (Scheme 2; Figure 2 and Figure 3, Table 1 and Appendix A). The obtained hydrolytic products were afterwards deliberately synthesized and fully characterized (see experimental). It is interesting to note that, notwithstanding the similar constituent moieties, several differences can be evidenced between the neutral amidodithiophosphonato nickel complexes [Ni(BzNH-adtp)_2_] and [Ni(PhBuNH-adtp)_2_] and the corresponding Ni-dithiophosphonate ammonium salts (BzNH_3_)_2_[Ni(dtp)_2_] and (PhBuNH_3_)_2_[Ni(dtp)_2_], mainly regarding the melting points and the IR and NMR signals relative to the ammonium -NH_3_^+^ groups (Appendix A). 

The asymmetric unit of the (BzNH_3_)_2_[Ni(dtp)_2_]·2H_2_O comprises half a molecule, with a Ni^II^ ion lying about a crystallographic inversion center, one dithiophosphonato ligand (ArPOS_2_)^−^, one benzylammonium cation, and one water molecule. The metal center is tetracoordinated in a square-planar geometry by four sulphur atoms belonging to two isobidentate ligands with Ni–S bond lengths of 2.2255(4) and 2.2232(4) Å, respectively, and an S–Ni–S angle of 87.92(2)°. The P–S1 and P–S2 bond lengths show very similar values [2.0409(5) and 2.0387(5) Å respectively], indicating an electron delocalization over the whole PS_2_^−^ fragment and an S–P–S angle of 98.39(2)°; the P–O bond exhibits a length of 1.5094(10) Å (Table 1). 

The structure of (PhBuNH_3_)_2_[Ni(dtp)_2_] (Figure 3) displays four nickel metal centers in the asymmetric unit, each sitting on a special position with its occupancy necessarily set to 0.5. One half of each metal complex is crystallographically unique, with the other chelated ligand being generated through the symmetry of the space group. There are also four crystallographically unique protonated organic amine counter ions, three of which are disordered (Appendix A). The nickel coordination closely resembles that found in (BzNH_3_)_2_[Ni(dtp)_2_]·2H_2_O (see above) with similar bond lengths and angles (Table 1), and an average P–O bond length of 1.511(2) Å.

It is interesting to note that the bond lengths and angles of the –POS_2_ moiety are comparable with those found in similar neutral and anionic dithiophosphonato nickel complexes described as bearing either a P=O or P–OH bond [7,8,9,10,33,34,35,36,37,38,39,40]. It is therefore very difficult (from comparison of the crystallographic P–O and/or P–S bond lengths alone) [7,8,9,10,33,34,35,36,37,38,39,40], to discriminate between purely single and double P–O bonds in this class of compounds, or to confidently assign the negative charge on either the oxygen or on the sulphur atoms. Additionally, the [ArPOS_2_]^−^ fragments are often engaged in strong H-bonds with counterions, which affects the bond lengths between the atoms involved. A better understanding of the nature of the P–O bond (and its charge distribution) may be gleaned if the bond lengths involved in the –POS_2_ fragment are considered together. The correlation reported in Figure 4 suggests that purely double P=O bonds may only be found on P(O)S_2_ fragments not directly bearing a negative charge (empty green circles, CCDC ref-codes IDUNEC, NEYLUA, NEYMAH, and YABQEZ) [7,8,9,10,33,34,35,36,37,38,39,40]. Similarly, pure single P–O bonds are detected in neutral O-alkyl-dithiophosphonates (**III** in Scheme 1, yellow circles in Figure 4). All the fragments bearing a negative charge fall in the same area, notwithstanding the attributions, as single P–OH (blue squares) or double P=O bonds (full green circles) reported for the deposited structures. Compounds (BzNH_3_)_2_[Ni(dtp)_2_]·2H_2_O (black triangle in Figure 4) and (PhBuNH_3_)_2_[Ni(dtp)_2_] (red triangle in Figure 4) lie in the same region as the anionic fragments. It is worth noting the three blue squares lying in the same region as the yellow dots: these data refer to structures (IKOSUX and LIFGAJ) [10,33,34,35,36,37,38,39,40] containing single P–OH bonds in neutral fragments, thus confirming the proposed correlation. 

The crystal packing of both (BzNH_3_)_2_[Ni(dtp)_2_]·2H_2_O and (PhBuNH_3_)_2_[Ni(dtp)_2_] are mainly governed by the strong H-bonds involving the anionic complexes, the ammonium BzNH_3_^+^ and PhBuNH_3_^+^ cations and the crystallized water molecules in (BzNH_3_)_2_[Ni(dtp)_2_]·2H_2_O (Appendix A). 

An attempt at crystallizing (PhBuNH_3_)(PhBuNH-adtp) in toluene yielded a few crystals of the salt (PhBuNH_3_)_2_[(ArPS_2_)_2_O] (Figure 5 and Appendix A; Table 1, Appendix A) containing the bis(4-methoxyphenyl)tetrathiodiphosphonate anion [(ArPS_2_)_2_O]^2−^ counterbalanced by two 4-phenylbutylammonium cations. The structure was solved in the space group P-1, and the main structural and refinement parameters are reported as Appendix A. There is a large extent of disorder in the crystal. The disordered atoms were modelled and refined over two or four positions using a combination of thermal and geometric parameter restraints and/or constraints where necessary (see Experimental).

Figure 5 shows the ellipsoid plot (50% probability level) and numbering scheme of one [(ArPS_2_)_2_O]^2−^ anion and one PhBuNH_3_^+^ cation; the complete content of the asymmetric unit with disorder shown is reported in Appendix A. Bond lengths and angles in the (S_2_P–O–PS_2_)^2−^ fragment are similar to those previously reported for the thionated(naphthalene-1,8-diyl)bis(phosphonic) acid monoanhydride [41]. The anions interact with ammonium cations through strong N–H···S hydrogen bonds (Appendix A). 

The isolation of compound (PhBuNH_3_)_2_[(ArPS_2_)_2_O] confirms that the hydrolytic process involves the cleavage of P–N bond in amidodithiophosphonates, while retaining the P–S, as already observed in (BzNH_3_)_2_[Ni(dtp)_2_]·2H_2_O and (PhBuNH_3_)_2_[Ni(dtp)_2_]. 

### 2.1. Antibacterial Activity

A set of different tests were performed to evaluate the antimicrobial activity of amidodithiophosphonate salts (^i^PrNH_3_)(^i^PrNH-adtp), (BzNH_3_)(BzNH-adtp), and (PhBuNH_3_)(PhBuNH-adtp) and the relevant nickel(II) complexes [Ni(^i^PrNH-adtp)_2_], [Ni(BzNH-adtp)_2_] and [Ni(PhBuNH-adtp)_2_] against different Gram-positive and Gram-negative bacterial species, both in planktonic and in sessile life. In this context, microbial species described as being commensals or pathogens in humans were assayed, namely *Staphylococcus aureus, Staphylococcus haemolyticus, Escherichia coli,* and two strains of *Pseudomonas aeruginosa*, *PA-01* and *PA-02*, that showed a different susceptibility pattern to disinfectants [42]. In addition, three different clinical isolates of *Candida spp*. were assayed, namely *Candida albicans*, *Candida kruseii*, and *Candida glabrata*. The antimicrobial activity of the ligand salts and nickel complexes was measured by the Agar diffusion method against the mentioned strains. These tests revealed that, while none of the tested Gram-negative bacteria or fungi were sensitive to any of the compounds examined, complexes [Ni(BzNH-adtp)_2_] and [Ni(PhBuNH-adtp)_2_] are active against both *Staphylococcus spp*. (Appendix A). In particular, a growth inhibition (Ø) of 12 and 8 mm was exerted on *S. aureus* and of 17 and 15 mm against *S. haemolyticus* by [Ni(BzNH-adtp)_2_] and [Ni(PhBuNH-adtp)_2_], respectively (Appendix A). The fact that there was activity against Gram-positive bacteria and not against Gram-negative may be related to the increased difficulty of these compounds to penetrate the cell wall of the Gram-negatives [43,44,45]. Notably, no inhibitory activity towards *S. aureus* and *S. haemolyticus* was observed for the complexes’ ligand precursors (BzNH_3_)(BzNH-adtp), (PhBuNH_3_)(PhBuNH-adtp), and NiCl_2_·6H_2_O, showing that the coordination compounds are responsible for the antimicrobial activity. In contrast, the inability of [Ni(^i^PrNH-adtp)_2_] to inhibit bacterial growth suggests that the tendency towards hydrolysis of the complexes [Ni(BzNH-adtp)_2_] and [Ni(PhBuNH-adtp)_2_] could play an important role in their antimicrobial activity. The tendency to hydrolysis could be tentatively ascribed to the different nature of the alkyl/aryl amine substituents, as evidenced by the slight elongation of the P–N bond on passing from [Ni(^i^PrNH-adtp)_2_] to [Ni(PhEtNH-adtp)_2_] (1.619(5) and 1.641(4) Å, respectively) [3,31,32]. 

Minimum inhibitory concentration (MIC) represents the lowest concentration of an antimicrobial that inhibits the visible growth of a microorganism after an appropriate incubation time. Evaluating the MIC confirmed the activity against *Staphylococci*.

Both [Ni(BzNH-adtp)_2_] and [Ni(PhBuNH-adtp)_2_] were observed to inhibit the growth of *S. aureus* up to just a 2-fold dilution of the stock solution (200 μg/mL; 5.95·10^−4^ and 5.00·10^−4^ M respectively). *S. haemolyticus* showed a MIC up to a 32-fold dilution in the case of [Ni(BzNH-adtp)_2_], corresponding to 6.25 μg/mL (1.56·10^−5^ M). In contrast, [Ni(PhBuNH-adtp)_2_] lost the ability to inhibit the bacterial growth after dilution (MIC > 100 μg/mL). Moreover, the bactericidal activity was assessed by evaluating the minimum bactericidal concentration (MBC), i.e., the lowest concentration of an antimicrobial required to kill a particular bacterium life in suspension (planktonic status). This approach is established when the substance under investigation can inactivate bacterial contamination in a fluid, such as water, saliva and urine. Neither [Ni(BzNH-adtp)_2_] nor [Ni(PhBuNH-adtp)_2_] showed any bactericidal activity (MBC > 100 μg/mL; Table 2) against these strains. Microorganisms living within a structured biofilm cause many human infections. Such a sessile structure is generally more resistant to various antimicrobial treatments [28]. For this reason, we measured the influence of complexes [Ni(BzNH-adtp)_2_] and [Ni(PhBuNH-adtp)_2_] on biofilm formation by evaluating the minimum concentration required to inhibit the formation of the biofilm in vitro, i.e., the minimum biofilm inhibitory concentration (MBIC). Both complexes showed the ability to inhibit the biofilm growth, however they required relatively high concentrations in the case of *S. aureus* (MBIC = 100 μg/mL). Notably, for both complexes, lower MBICs were observed for the biofilms of *S. haemolyticus* (MBIC = 50.0 and 12.5 μg/mL for [Ni(BzNH-adtp)_2_] and [Ni(PhBuNH-adtp)_2_], respectively, corresponding to 1.49·10^−4^ and 3.12·10^−5^ M). For the sake of comparison, MIC, MBC, and MBIC measurements were carried out on NiCl_2_·6H_2_O under the same experimental conditions, on the same strains of microrganisms. As expected, nickel chloride did not show any antibacterial activity towards *S. aureus* and *S. haemolyticus* (Appendix A). 

### 2.2. DFT Calculations

Following recent studies on different complexes featuring chalcogen donors [18,46,47,48,49], the electronic structures of salts (^i^PrNH_3_)(^i^PrNH-adtp), (BzNH_3_)(BzNH-adtp), and (PhBuNH_3_)(PhBuNH-adtp), and the corresponding Ni^II^ complexes were investigated by theoretical calculations carried out at the density functional theory (DFT) [50] in order to theorize why complexes [Ni(BzNH-adtp)_2_] and [Ni(PhBuNH-adtp)_2_] undergo hydrolysis and try to explain the different antimicrobial activity determined between these complexes and the analogous [Ni(^i^PrNH-adtp)_2_] and the free amidodithiophosphonate salts. DFT calculations were carried out on the starting amines ^i^PrNH_2_, BzNH_2_, PhBuNH_2_, the corresponding ammonium cations, the relevant amidodithiophosphonate anions (^i^PrNH-adtp)^−^, (BzNH-adtp)^−^, and (PhBuNH-adtp)^−^, and the complexes [Ni(^i^PrNH-adtp)_2_], [Ni(BzNH-adtp)_2_] and [Ni(PhBuNH-adtp)_2_]. As expected, all the starting amines feature a Kohn–Sham (KS) HOMO corresponding to the lone pair (LP) of electrons localized on the nitrogen atoms (with eigenvalues of −6.612, 6.898, and −6.579 eV, for ^i^PrNH_2_, BzNH_2_, PhBuNH_2_, respectively). This LP is therefore available to react with Lawesson’s Reagent, which features a positive charge on the P atom (*Q* = 0.878 |e|) accompanied by a low energy (−2.075 eV) KS-LUMO, antibonding in nature with respect to the P–S bonds of the P_2_S_2_ 1,3-dithia-2,4-diphosphetane ring. The (^i^PrNH-adtp)^−^, (BzNH-adtp)^−^, and (PhBuNH-adtp)^−^ anions feature strongly polarized P–N bonds (NBO charges *Q*_P_: 1.269, 1.262, 1.264; *Q*_N_: −1.012, −1.002, −1.000 |e|, respectively). This results in Wiberg bond indices (WBI) sensibly lower than unity (WBI_PN_ = 0.721, 0.697, 0.707, respectively), reflected in optimized P–N bond lengths (*d*_PN_) in the range between 1.741 and 1.749 Å. All the anions feature virtual MOs antibonding with respect to the P–N bonds. Moving from the anions to the corresponding Ni^II^ complexes, the P–N bonds are slightly strengthened (WBI_PN_ 0.791–0.828; *d*_PN_ 1.674–1.679 Å). In analogy to the corresponding ligands, these complexes feature low-lying virtual MOs, antibonding with respect to the P–N bonds. In addition, the charge on the P atoms, and therefore the polarization of the P–N bonds, increases, thus indicating a large electrophilic character of the P atoms (*Q*_P_ = 1.365, 1.361, 1.361; *Q*_N_ = −1.010, −1.006, −1.007 |e| for [Ni(^i^PrNH-adtp)_2_], [Ni(BzNH-adtp)_2_] and [Ni(PhBuNH-adtp)_2_], respectively). As a result, the eigenvalues of the virtual antibonding MOs (with respect to the P–N bonds) are more stable than those of the corresponding free anionic ligands (0.672, −0.211, 0.712 eV, respectively) and result in an increased electrophilic character of the P atom. Therefore, the susceptibility to the hydrolysis of related amidodithiophosphonato Ni^II^ complexes is expected to be larger than that of the starting ligands. 

These results are in line with the hypothesis that the antibacterial activity could be related to the hydrolytic process. The subsequent cleavage of the P–N bond in the neutral [Ni(adtp)_2_] complexes in turn leads to the formation of the complexes (BzNH_3_)_2_[Ni(dtp)_2_] and (PhBuNH_3_)_2_[Ni(dtp)_2_] that have been isolated and structurally characterized (Scheme 2; Figure 2 and Figure 3, Table 1 and Table 2). This hypothesis would also be consistent with the lack of antibacterial activity determined for [Ni(^i^PrNH-adtp)_2_], which has proved experimentally to be less prone to hydrolysis, as confirmed by the stability in solution and isolation in the solid state [3]. It is interesting to note that no virtual MOs (antibonding with respect to the P–S bonds) can be found at energies close to, or lower than, those of the aforementioned antibonding P–N MOs, either in the free R-adtp^−^ anions or in the corresponding [Ni(R-adtp)_2_] complexes. This indicates that hydrolysis of the compounds should be expected to occur through P–N bond breaking and dithiophosphonate anion formation, causing some doubt regarding the previously hypothesized emission of dihydrogen sulfide as the first step of the hydrolysis [51]. The hydrolysis of P–N was also confirmed by the isolation of a few crystals of the 4-phenylbutylammonium salt of bis(4-methoxyphenyl)-tetrathiodiphosphonate (PhBuNH_3_)_2_[(ArPS_2_)_2_O] (Figure 5) during an attempt at crystallizing (PhBuNH_3_)(PhBuNH-adtp). A similar salt was hypothesized as the intermediate in the in situ formation of a mixed cymene-ferrocenylphosphonodithiolate ruthenium complex, obtained by the hydrolysis of 2,4-diferrocenyl-1,3-dithiadiphosphetane 2,4-disulfide in the presence of ammonium hydroxide [35].

## 3. Materials and Methods

Starting materials and solvents were purchased from commercial sources TCI (Tokio, Japan) and Aldrich (Darmstadt, Germany) and, when necessary, the solvents have been distilled and dried according to the standard literature techniques. Melting point measurements were determined in capillaries, using melting point apparatus BUCHI M-560 (30–240 °C, Flawil, Svizzera). Elemental analyses were performed with an EA1108 CHNS-O Fisons instrument (Thermo Fisons, Okehampton, EX20 1UB, UK). ^1^H and ^31^P NMR measurements were carried out at 25 °C using a Bruker Avance 300 MHz (7.05 T, Billerica, MA, USA) spectrometer at operating frequencies of 300.13 and 121.49 MHz, respectively. Chemical shifts for ^1^H-NMR are reported in parts per million (ppm), calibrated to the residual solvent peak set, with coupling constants *J* reported in Hertz (Hz). Chemical shifts for ^31^P NMR are reported in parts per million (ppm), calibrated to the external reference TPP 48.5 mM in acetone-*d*_6_. Infrared (IR) spectra were recorded on a Thermo Nicolet 5700 FT-IR spectrophotometer (Waltham, MA, USA) using KBr pellets and reported in wavenumbers (cm^−1^).

Single-crystal X-ray diffraction data were collected at 100 K on a Rigaku FRE+ equipped with VHF Varimax confocal mirrors and an AFC12 goniometer (Tokio, Japan) and HyPix 6000 detector diffractometer (Tokio, Japan) [52]. The structures were solved with the ShelXT [53] structure solution program using the Intrinsic Phasing solution method, using Olex2 [54] as the graphical interface. The model was refined with version 2018/3 of ShelXL [55] using Least Squares minimization. All hydrogen atoms were added in calculated positions and refined in riding positions relative to the parent atom. CCDC deposition numbers: 1944063–1944065.

### 3.1. Theoretical Calculations

Quantum-mechanical calculations were carried out at density functional theory (DFT) [50] level (mPW1PW functional) [56] with the Gaussian 16 (rev B.01, Gaussian Inc., Wallingford, CT, USA) [57] commercial suite of computational software. All calculations were performed by adopting the def2-SVP [58] basis sets for all atomic species. The calculations were carried out on the amines ^i^PrNH_2_, BzNH_2_, PhBuNH_2_, the corresponding ammonium cations, the relevant amidodithiophosphonate anions (^i^PrNH-adtp)^−^, (BzNH-adtp)^−^, and (PhBuNH-adtp)^−^, and nickel complexes [Ni(^i^PrNH-adtp)_2_], [Ni(BzNH-adtp)_2_] and [Ni(PhBuNH-adtp)_2_]. For all the investigated compounds the geometries were optimized, starting from structural data when available, and symmetrized to achieve the highest possible point group. For all the neutral, anionic, and cationic species tight convergence criteria were adopted (maximum force 1.5·10^−5^ Ha Bohr^−1^, RMS force 1.0·10^−5^ Ha Bohr^−1^, maximum displacement 6.0·10^−5^ Å, and RMS displacement 4.0·10^−5^ Å). The nature of the energy minima at the optimized geometries were verified by a vibrational analysis, computed by determining the second derivatives of the energy with respect to the Cartesian atomic coordinates and subsequently transforming to mass-weighted coordinates. Natural bonding orbitals [59], natural charges and Wiberg bond indices [60] were calculated at the optimized geometries. The programs Chemissian [61], GaussView 6.0 [62], and Molden 5.9 [63] were used to analyze optimized geometries and Kohn–Sham molecular orbitals.

### 3.2. Microbiological Assays

The following species were used: (i) Gram-positive bacteria, *Staphylococcus aureus* ATCC 6538 (American Type Culture Collection), *Staphylococcus haemolyticus* clinical isolate NC1; (ii) Gram-negative bacteria, *Escherichia coli* ATCC 7075, and two strains of *Pseudomonas aeruginosa: P. aeruginosa ATCC 15442 (PA-1)* recommended for disinfectant testing by official methods was used as the high biocide-resistant strain; *P. aeruginosa ATCC 27853 (PA-2)* was used as the susceptible disinfectant strain. In addition, the clinical isolates of *Candida spp*. were assayed: *C. albicans BF1, C. kruseii BF2* and *C. glabrata BF3.* In vitro susceptibility testing was carried out on compounds (^i^PrNH_3_)(^i^PrNH-adtp), (BzNH_3_)(BzNH-adtp), (PhBuNH_3_)(PhBuNH-adtp), NiCl_2_·H_2_O, and complexes [Ni(^i^PrNH-adtp)_2_], [Ni(BzNH-adtp)_2_] and [Ni(PhBuNH-adtp)_2_] using: (a) the agar diffusion method, (b) minimum inhibitory concentration (MIC) and (c) minimum bactericidal concentration (MBC), determined in accordance with the Clinical Laboratory Standard Institute CLSI [64,65]. The Agar diffusion method was performed by using the Kirby–Bauer procedure [66], and it was employed to reveal the entire antimicrobial susceptibility profile for the examined compounds. 1·10^7^ cells/mL were inoculated onto the surface of an agar plate containing one of the subsequent bacterial growth agar media manufactured by Microbiol (Uta, Italy): (i) Muller–Hinton agar was used for aerobic bacteria, (ii) Fungi on Sabouraud agar. Each agar plate contained a central circular cavity, Ø = 10 mm, able to contain 50 μL volume of the solution of the examined compounds (C: (^i^PrNH_3_)(^i^PrNH-adtp) = 1.28·10^−3^; (BzNH_3_)(BzNH-adtp) = 7.93·10^−4^; (PhBuNH_3_)(PhBuNH-adtp) = 6.17·10^−4^, NiCl_2_·6H_2_O = 5.00·10^−4^, [Ni(^i^PrNH-adtp)_2_] = 8.33·10^−4^, [Ni(BzNH-adtp)_2_] = 5.95·10^−4^ and [Ni(PhBuNH-adtp)_2_] = 5.00·10^−4^ M in 1/10 *v/v* DMSO/water). The antimicrobial activity was expressed as mm of inhibition diameter around the cavity, after the microbial growth at 37 °C. All experiments were performed in triplicate and the values shown are reported as the average ± standard deviation of the inhibition diameter. MIC and MBC were determined only against the microbial strains susceptible to Kirby–Bauer assay, according to the micro-broth dilution method [67,68], by using a 1/2 serial dilution, from 100 to 0.20 μg/mL of the compounds under study in the previously described liquid growth mediums (C: (^i^PrNH_3_)(^i^PrNH-adtp) = 2.56·10^−6^, (BzNH_3_)(BzNH-adtp) = 1.59·10^−6^, (PhBuNH_3_)(PhBuNH-adtp) = 1.23·10^−6^, NiCl_2_·6H_2_O = 1.00·10^−6^, Ni(^i^PrNH-adtp)_2_ = 1.67·10^−6^, Ni(BzNH-adtp)_2_ = 1.19·10^−6^ and Ni(PhBuNH-adtp)_2_ = 1.00·10^−6^ M). For the biofilm evaluation, we used the protocol described by Montana University’s Center for Biofilm Engineering [69]. A microplate containing serial concentrations of the compound, inoculated with the bacterial strains as previously described for MIC and MBC evaluation, was incubated at 37 °C for 48 h, to permit the biofilm formation. The plate samples were subsequently washed three times with phosphate-buffered saline GIBCO^®^PBS (Thermo Fisher, Waltham, MA, USA) to eliminate planktonic cells; thus, the biofilm was stained with 100 μL of 0.1% *w/v* of crystal violet solution (Microbial, Uta, Italy) for 10 min at 25 °C; after three washes with PBS solution, 200 μL of 30% *v/v* acetic acid was added in every well to solubilize the dye from the bacterial biomass. The biofilm amount was measured with a plate reader spectrophotometer (SLT-Spectra II, SLT Tecan Instruments, Männedorf, Switzerland) at 620 nm.

### 3.3. Synthesis

(^i^PrNH_3_)(^i^PrNH-adtp), (BzNH_3_)(BzNH-adtp) were synthesized as reported previously [3,31,32,70].

#### 3.3.1. Synthesis of (PhBuNH_3_)(PhBuNH-adtp)

Lawesson’s Reagent (0.972 g; 2.40 × 10^−3^ mol) was suspended in dry toluene (25 mL) in a sealed flask and stirred under nitrogen atmosphere. After few minutes 4-phenylbutylamine (PhBuNH_2_) (1.90 mL; 1.20 × 10^−2^ mol) was added dropwise. The reaction mixture was stirred vigorously for 5 h and cooled at −10 °C; the precipitate was then filtered under reduced pressure and washed several times with cold toluene. The solid was then suspended in diethyl ether for 30 min, filtered and dried under vacuum (0.427 g; Y = 18%). M.p. 143 °C (dec.) Elemental analysis calculated (%) for C_27_H_37_N_2_OPS_2_: C 64.77; H 7.45; N 5.59; S 12.81; found C 65.01; H 7.53; N 5.61, S 12.65. FT-IR (KBr, 4000–400 cm^−1^, Appendix A): 2928br, 1594s, 1569ms, 1492s, 1462ms, 1403w, 1301ms, 1253s, 1173m, 1137w, 1105s, 1028ms, 989w, 855vs, 800m, 749m, 717w, 697s, 658vs ν_asym_(P–S), 624s, 551ms ν_sym_(P–S), 516m, 439m cm^−1^. ^1^H-NMR (300 MHz, DMSO-*d*_6_, Appendix A) δ(ppm) 8.01–7.87 (m, 4H), 7.36–7.24 (m, 4H), 7.24–7.12 (m, 9H), 6.71 (d, *J* = 8.3 Hz, 4H), 3.73 (s, 6H), 2.81 (t, *J* = 7.2 Hz, 4H), 2.58 (t, *J* = 7.1 Hz, 4H), 1.71–1.45 (m, 8H). ^31^P{^1^H} NMR (121 MHz, DMSO-*d*_6_) δ(ppm) 93.1 (s).

#### 3.3.2. Synthesis of [Ni(BzNH-adtp)_2_]

NiCl_2_·6H_2_O (12.0 mg; 5.05 × 10^−5^ mol) was dissolved in MeOH (5 mL) and added dropwise to a suspension of (BzNH_3_)(BzNH-adtp) (42.2 mg; 1.01 × 10^−4^ mol) in MeCN (5 mL). The reaction mixture was then stirred at room temperature for 8 h. The purple solid was filtered and washed with acetonitrile (12.3 mg; Y = 36%) M.p. 162 °C (dec.) Elemental analysis calculated (%) for C_28_H_30_N_2_NiO_2_P_2_S_4_: C 49.79; H 4.48; N 4.15; S 18.99; found C 49.81; H 4.55; N 4.21, S 18.75. FT-IR (KBr, 4000–400 cm^−1^, Appendix A): 3251m ν(N-H), 3026w, 2926w, 2837w, 1951ms, 1570w, 1498ms, 1462m, 1454m, 1441w, 1406m, 1306mw, 1294m, 1428vs, 1209w, 1178m, 1113s, 1061s, 1022m, 968mw, 916w, 872m, 823ms, 816mw, 800m, 744ms, 688ms, 658m ν_asym_(P–S), 636w, 609m, 588w, 557ms ν_sym_(P–S), 523w, 496w, 422w cm^−1^. ^1^H-NMR (300 MHz, CDCl_3_, Appendix A) δ 7.89 (dd, *J* = 12.5, 8.6 Hz, 2H), 7.31–7.09 (m, 10H), 6.81–6.76 (m, 2H), 3.73 (s, 3H), 2.78 (t, *J* = 7.4 Hz, 2H), 2.68 (br, 1H, NH), 2.64–2.54 (m, 4H), 2.45 (t, *J* = 7.6 Hz, 2H), 1.66–1.40 (m, 6H), 1.35–1.26 (m, 2H). ^31^P{^1^H} NMR (121 MHz, DMSO-*d*_6_) δ 75.8 (s).

#### 3.3.3. Synthesis of (BzNH_3_)_2_[Ni(dtp)_2_]

NiCl_2_·6H_2_O (12.2 mg; 5.13 × 10^−5^ mol) was dissolved in H_2_O (5 mL) and added dropwise to a suspension of (BzNH_3_)(BzNH-adtp) (41.9 mg; 1.00 × 10^−4^ mol) in MeCN (5 mL). The purple solution was stirred at room temperature for 1 h, filtered and left to rest for overnight (20.4 mg; Y = 57%). M.p. > 240 °C (dec). Elemental analysis calculated (%) for C_28_H_38_N_2_NiO_6_P_2_S_4_: C 44.99; H 5.15; N 3.75; S 17.16; found C 50.06; H 5.23; N 3.77, S 17.05. FT-IR (KBr, 4000–400 cm^−1^, Appendix A): 3473m ν(N-H), 3034w, 2603w, 1620mw, 1595ms, 1570mw, 1497ms, 1456m, 1441w, 1402w, 1385w, 1296w, 1292m, 1252s, 1213w, 1174m, 1119s, 1076vs, 1026ms, 922w, 829m, 812w, 798m, 741m, 694m, 656m ν_asym_(P–S), 627w, 596ms, 548ms ν_sym_(P–S), 523w cm^−1^. 1H-NMR (600 MHz, DMSO-*d*_6_, Appendix A) δ 8.25 (dd, *J* = 11.5, 8.6 Hz, 4H), 8.21 (br, NH, 6H), 7.53–7.46 (m, *J* = 7.2 Hz, 4H), 7.46–7.41 (m, 4H), 7.41–7.36 (m, 2H), 6.92 (d, *J* = 7.0 Hz, 4H), 4.07 (s, CH2, 4H), 3.80 (s, OCH_3_, 6H).

#### 3.3.4. Synthesis of [Ni(PhBuNH-adtp)_2_]

NiCl_2_·6H_2_O (12.0 mg; 5.05 × 10^−5^ mol) was dissolved in MeOH (5 mL) and added dropwise to a suspension of (PhBuNH_3_)(PhBuNH-adtp) (50.1 mg; 1.00 × 10^−4^ mol) in MeCN (5 mL). The reaction mixture was then stirred at room temperature for 8 h. The purple solid was filtered and washed with chloroform (14.0 mg; Y = 37%). M.p. 147 °C. Elemental analysis calculated (%) for C_34_H_42_N_2_NiO_2_P_2_S_4_: C 53.76; H 5.75; N 3.62; S 16.88; found C 53.81; H 5.85; N 3.61, S 16.75. FT-IR (KBr, 4000–400 cm^−1^, Appendix A): 3261m ν(N-H), 3022w, 2937w, 2854w, 1591ms, 1568m, 1498ms, 1454m, 1439m, 1404m, 1306mw, 1292m, 1252vs, 1176ms, 1115s, 1080s, 1051m, 1024ms, 970m, 920m, 868m, 825ms, 798m, 750m, 735m, 698m, 660m ν_asym_(P–S), 633mw, 606ms, 573w, 548s ν_sym_(P–S), 525m, 469w, 401w cm^−1^. ^1^H-NMR (300 MHz, CDCl_3_, Appendix A) δ(ppm) 7.90 (dd, 4H), 7.33–7.11 (m), 6.97 (d, *J* = 7.1 Hz, 4H), 3.86 (s, 6H, OCH_3_), 3.17–2.99 (m, 4H, -CH_2_-NH), 2.97–2.83 (m, 2H, NH), 2.62 (d, *J* = 7.1 Hz, 4H, -CH_2_-Ph), 1.77–1.61 (m). ^31^P{^1^H} NMR (121 MHz, DMSO-*d*_6_) δ(ppm) 76.2 (s).

#### 3.3.5. Synthesis of (PhBuNH_3_)_2_[Ni(dtp)_2_]

NiCl_2_·6H_2_O (12.1 mg; 5.09 × 10^−5^ mol) was dissolved in H_2_O (5 mL) and added dropwise to a suspension of (PhBuNH_3_)(PhBuNH-adtp) (50.5 mg; 1.01 × 10^−4^ mol) in MeCN (5 mL). The purple solution was stirred at room temperature for 1 h, filtered and left to rest overnight. The solid was then filtered and recrystallized from chloroform (24.8 mg; Y = 62%). M.p. > 240 °C (dec). Elemental analysis calculated (%) for C_34_H_46_N_2_NiO_4_P_2_S_4_: C 51.33; H 5.83; N 3.52; S 16.12; found C 51.41; H 5.86; N 3.54, S 16.15. FT-IR (KBr, 4000–400 cm^−1^, Appendix A): 3248m ν(N-H), 3024vs, 2972vs, 2931vs, 2858s, 2607w, 2515w, 1953s, 1566m, 1498s, 1456m, 1454m, 1404mw, 1300m, 1296m, 1254s, 1180m, 1115s, 1063ms, 1022m, 970w, 920mw, 870w, 835m, 800m, 742m, 698m, 654m ν_asym_(P–S), 629w, 607ms, 544ms ν_sym_(P–S), 523w, 467mw, 407m cm^−1^. ^1^H-NMR (600 MHz, DMSO-*d*_6_, Appendix A) δ 8.30–8.20 (m, 4H), 7.80 (s, 6H), 7.28 (t, *J* = 7.3 Hz, 4H), 7.23–7.14 (m, 6H), 6.94 (d, *J* = 7.5 Hz, 4H), 3.80 (s, OCH_3_, 6H), 2.82 (m, 4H), 2.60 (t, *J* = 7.1 Hz, 4H), 1.67–1.51 (m, 8H).

#### 3.3.6. Synthesis of (PhBuNH_3_)_2_[(ArPS_2_)_2_O]

Few crystals of bis(4-methoxyphenyl)-tetrathiodiphosphonate (PhBuNH_3_)_2_[(ArPS_2_)_2_O] were obtained during an attempt at crystallizing (PhBuNH_3_)(PhBuNH-adtp) by slow diffusion of ethyl ether in a toluene solution of the salt. The very exiguous amount of compound prevented us from further characterizations.

## 4. Conclusions

The reaction between Lawesson’s Reagent (LR) and isopropylamine (^i^PrNH_2_), benzylamine (BzNH_2_), and 4-phenylbutylamine (PhBuNH_2_) in toluene gave rise to the corresponding amidodithiophosphonate ammonium salts (^i^PrNH_3_)(^i^PrNH-adtp), (BzNH_3_)(BzNH-adtp), and (PhBuNH_3_)(PhBuNH-adtp) that were reacted with nickel chloride hexahydrate, yielding the corresponding amidodithiophosphonato complexes [Ni(^i^PrNH-adtp)_2_], [Ni(BzNH-adtp)_2_] and [Ni(PhBuNH-adtp)_2_]. All the compounds were tested against a library of bacteria and fungi of clinical importance belonging to the genera *Staphylococcus*, *Escherichia*, and *Pseudomonas*, and *Candida*, but only the complexes [Ni(BzNH-adtp)_2_] and [Ni(PhBuNH-adtp)_2_] demonstrated some antimicrobial activity that was tentatively ascribed to their tendency towards hydrolysis. Theoretical and experimental results evidenced that [Ni(BzNH-adtp)_2_] and [Ni(PhBuNH-adtp)_2_] undergo hydrolysis and that during the hydrolytic process a cleavage of the polarized P–N bond occurs with consequent formation of a P–O bond and retaining of the two P–S bonds in the amidodithiophosphonate moiety. Even if hydrolysis was proven to occur both in the amidodithiophosphonate salts and in the corresponding nickel complexes, an increased polarization of the P–N bond was calculated for the latter, suggesting a higher tendency to undergo hydrolysis. The antibacterial inactivity of the salts can be tentatively explained by taking into account their high hydrophilicity associated with their ionic nature, which circumvents the penetration of the cellular membrane. On the contrary, the neutral complexes [Ni(BzNH-adtp)_2_] and [Ni(PhBuNH-adtp)_2_] can pass the cellular membrane and thus exploit their activity. The inactivity of the analogous [Ni(^i^PrNH-adtp)_2_], can be explained, taking into account its higher resistance to hydrolysis, demonstrated by its higher stability both in solution and in the solid state, also confirmed by a P–N bond that is slightly shorter than those determined for analogous phenyl-alkyl-amidodithiophosphonato complexes. Further studies are ongoing in order to better understand the role of the alkyl-aryl substituents of the amines in the final amidodithiophosphonato complexes.

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
