# Peer review of "Antibacterial Activity of Amidodithiophosphonato Nickel(II) Complexes: An Experimental and Theoretical Approach"

_molecules, 2020, doi:10.3390/molecules25092052_

Round 1

Reviewer 1 Report

This article is well written but can be published after a minor revision, as follows: 

Abstract: 

    -It must be specified which abbreviation corresponds to the compound (1); the abbreviation must be in chronological order, in the text of the abstract.You cannot start with (2) by omitting (1) in the abstract, even if in the chapter "Results" the abbreviation is correct and chronological.

 -The abbreviation LR for Lawesson's Reagent must appear both in abstract and   manuscript, there where the agent's name appears for the first time.

"3.3.1-3.3.5. Synthesis" subchapters: the introduction in the manuscript of the images from the FTIR and XRD analyses could better highlight the results.

In the context of the spread of COVID-19 generated by this highly contagious coronavirus, I wish all the authors much health, do not be affected in any way, and all of you overcome this difficult situation.

Reviewer 2 Report

Manuscript Review: Antibacterial Activity of Amidodithiophosphonato Nickel (II) Complexes: An Experimental and Theoretical Approach

To properly determine the antimicrobial activity, it is necessary to perform the study for the nickel salt that was used to obtain the complexes. The MIC tests for reference substances should be attached to the table to correctly determine the activity of the complexes tested. Including these studies in your work will change discussions about activity. Without these elements, the work cannot be evaluated.

Reviewer 3 Report

This is an interesting paper that shows antibacterial properties of the nickel(II) complexes, although the supplementary files including cif files are missing, which makes me difficult to give the decision.  Following points should be reconsidered for the next submission.

  • Figure 1 is really needed for the second line of page 2.
  • Scheme 2 is not easy to understand and should be described more clearly.
  • The structural parameters of (2H+)2[Ni(dtp)2]2-.2H2O and (3H+)2[Ni(dtp)2]2- should be discussed comparing with those [Ni(1-adtp)2] (ref. 2), which may give the useful information for the reason why [Ni(1-adtyp)2] do not show antimicrcrobial activity. The comparison may also support the DFT calculation results.

Round 2

Reviewer 2 Report

I made a review on March 18. The authors did not take my comments into account.
I attach the previous review below.
I can make a new review if the authors take into account my comments.

Manuscript Review: Antibacterial Activity of Amidodithiophosphonato Nickel (II) Complexes: An Experimental and Theoretical Approach

To properly determine the antimicrobial activity, it is necessary to perform the study for the nickel salt that was used to obtain the complexes. The MIC tests for reference substances should be attached to the table to correctly determine the activity of the complexes tested. Including these studies in your work will change discussions about activity. Without these elements, the work cannot be evaluated.

Reviewer 3 Report

The manuscript has been properly performed.  This paper is considered to be acceptable, although some corrections should be made for publication.  They are itemized below:

  • CCDC numbers should be given in the text for the crystallographically determined complexes (BzNH3)2[Ni(dtp)2].2H2O, (PhBuNH3)2[Ni(dtp)2], and (PhBuNH3)2[(ArPS2)2O]. (The cif check results might be needed in Supplementary Materials.)
  • In the section of “Synthesis” (Page 13): The authors say that suitable crystals of (BzNH3)2[Ni(dtp)2].2H2O and (PhBuNH3)2[Ni(dtp)2] were obtained by using the reactions of (BzNH3)(BzNH-adtp) or (PhBuNH3)(PhBuNH-adtp) (in MeCN) with NiCl2.6H2O (in H2O).  This description is different form the explanation in Page 5 (136th line), which shows that the crystals of (BzNH3)2[Ni(dtp)2].2H2O and (PhBuNH3)2[Ni(dtp)2] were obtained by recrystallizations of [Ni(BzNH-adtp)2] and [Ni(PhBuNH-adtp)2] from acetonitrile or chloroform solutions.
  • At 443th line of Page 13: “Suitable crystals for…” should be removed, because a similar description is given at 457th line of this page for the synthesis of (PhBuNH3)2[Ni(dtp)2].
  • The experimental condition to get crystals of (PhBuNH3)2[(ArPS2)2O] should be briefly described in the section of “Synthesis” if it is a new compound, because the characterization has been made using NMR spectrum.
  • At 260th line: “and” should be removed.
